# Hidden topological constellations and polyvalent charges in chiral nematic droplets

Gregor Posnjak[1], Simon Čopar[2] & Igor Muševič[1,2]

Topology has an increasingly important role in the physics of condensed matter, quantum systems, material science, photonics and biology, with spectacular realizations of topological concepts in liquid crystals. Here we report on long-lived hidden topological states in thermally quenched, chiral nematic droplets, formed from string-like, triangular and polyhedral constellations of monovalent and polyvalent singular point defects. These topological defects are regularly packed into a spherical liquid volume and stabilized by the elastic energy barrier due to the helical structure and confinement of the liquid crystal in the micro-sphere. We observe, for the first time, topological three-dimensional point defects of the quantized hedgehog charge $q = -2, -3$. These higher-charge defects act as ideal polyvalent artificial atoms, binding the defects into polyhedral constellations representing topological molecules.

[1] Condensed Matter Department, Jožef Stefan Institute, Jamova 39, 1000 Ljubljana, Slovenia. [2] Department of Physics, Faculty of Mathematics and Physics, University of Ljubljana, Jadranska 19, 1000 Ljubljana, Slovenia. Correspondence and requests for materials should be addressed to I.M. (email: igor.musevic@ijs.si).

Topology currently emerges as a major research theme in a number of subfields of physics, and many new topological phenomena have been observed in very different contexts. For example, nontrivial topology of the electronic structure of crystalline materials leads to exotic material properties, such as surface conductivity of topological insulators[1,2] and peculiar electronic structure of Weyl and Dirac semimetals[3–6]. Topologically important phenomena can nowadays be traced in condensed matter[1], quantum systems[7], material science[8], photonics[9], active matter[10–12] and even cell division processes in biology[13]. Understanding the topological properties of matter and applying topological concepts across various subfields of physics not only helps us understand the physics of these systems, but can inspire design of new materials with unusual material properties.

Liquid crystals (LCs) are well known for their richness of topological defects and phenomena, which are observed and analysed on the micrometre scale using optical methods. During the past decade, several spectacular realizations of topological concepts have been demonstrated in LCs, such as topological charge creation and manipulation in nematic liquid crystals (NLCs)[14–16], and fascinating defect motion in active NLCs[11,12]. By using laser tweezers and fluorescent confocal polarized microscopy (FCPM) it is possible to analyse the topological charges, knot and link the tensorial nematic ordering field[17–19] and experimentally prove fundamental theorems, such as the Poincaré-Hopf theorem[20,21].

Closely related to topology are hidden states of matter, which can be stabilized by topology and refer to states that are not accessible under equilibrium conditions, but can be created if the system is rapidly quenched[22]. Optically induced hidden states were reported in metallic glasses[23] and layered, quasi-two-dimensional electronic crystals[24] by applying strong laser pulses. In LCs, transient hidden states of topological defects are obtained by a rapid temperature or pressure quench across the isotropic-to-nematic phase transition. This quench generates a random constellation of topological defects via the Kibble-Zurek mechanism[25–27], consisting of mutually compensating topological charges, which are unstable and annihilate into vacuum. Topological defects of unit charge can be stabilized in LCs by colloidal inclusions, appearing as micrometre-scale, loop-like structural imperfections, accompanying spheres[17], fibres[14], handlebodies[20] and knotted particles[19]. In chiral nematic liquid crystals, topological monopoles of unit charge are stabilized by the spontaneous winding of the nematic orientational field, which forms skyrmion-like twisted three-dimensional (3D) structures called torons[28–30]. Topological defects of higher than unit charge have not been observed before in nematic LCs, as an equivalent number of smaller charges usually achieves a lower free energy due to a lower elastic distortion[31,32].

NLCs are characterized by spontaneous orientational ordering of long axes of rod-like molecules, which is combined with complete positional disorder of the centres of gravity of molecules. This orientation order is described with a tensorial orientational field, where the largest eigenvalue of this Q-tensor is pointing into a direction called the director, also referred to as a headless vector **n**. If the constituent molecules are chiral, the nematic phase spontaneously twists along a direction perpendicular to the director and forms twisted or chiral nematic liquid crystalline phase. When such a spontaneously twisted LC is confined into a sphere, which enforces perpendicular orientation of the LC molecules on the interface, the orientational field is frustrated and can form a variety of different topological states. It has recently been demonstrated experimentally that these states are rich in topological point defects[30], where the core of the defect is molten and the orientational order is strongly depressed. Theory also predicted the existence of linked and knotted loop

defects in chiral nematic droplets[33], but such topological states have not yet been confirmed.

Here we show that long-lived topological states in thermally quenched chiral nematic droplets are formed from string-like, triangular and polyhedral constellations of monovalent and polyvalent singular point defects. Topological defects are regularly packed into a spherical liquid volume and stabilized by the elastic energy barrier due to the helical structure and confinement of the LC in the micro-sphere. We observe for the first time the quantization of topological charge of 3D point defects in terms of a unit topological charge. In addition to the $q = -1$ hedgehogs, which are usually stable in LC droplets and colloids, we observe higher charges with a multiple hedgehog charge $q = -2$, and $-3$. These monster charges can be regarded as polyvalent artificial atoms, and are able to bind the surrounding unit charge defects into polyhedral constellations representing topological molecules.

## Results

**Quenched chiral nematic droplets**. To obtain droplets with various topological constellations of point defects we mix chiral LCs with chiral pitch in the range 6 to 12 μm into a viscous liquid, which promotes perpendicular anchoring to form droplets with diameters 10 to 20 μm (see Methods for details). After the droplets are formed, the sample is heated to the isotropic phase and cooled at a rate of several degrees per second back to the chiral nematic phase. This produces topologically complex and higher free energy director structures, which are stable for several days[33,34]. The complexity of the topological states in chiral nematic droplets depends on the chirality parameter $N = 2d/p_0$, where $d$ is the diameter of the droplet and $p_0$ is the intrinsic pitch of the chiral nematic LC. By far the most common are layered cholesteric structures with a single point defect, which is due to the topology of spherical confinement, but for $N > 2.5$ structures with more point defects can appear. The droplets are examined by wide-field optical and FCPM, and director fields are reconstructed from FCPM data using a recently developed method[30] (details are described in Methods).

In samples with a chirality $N \sim 2.5$–4 we obtain two $+1$ radial hedgehogs residing at the surface of the droplet and facing each other on a symmetry axis, as shown in Fig. 1a–h. Their topological charge amounts to $+2$ and another hyperbolic hedgehog with negative charge $-1$ appears in between, adding up the total topological charge inside the sphere to $+1$. This is because the Poincaré-Hopf theorem requires the net hedgehog charge $q$ of the director field $\mathbf{n}(r)$ with perpendicular surface anchoring along the closed bounding surface $S$ with genus $g$ to be equal to (up to sign) $q = \pm (1 - g)$ (refs 31,32). For a sphere with $g = 0$, having no handles, there should be a total hedgehog charge of $+1$, where we choose the sign according to the convention that the director field, represented as a vector field, points outwards from the surface of the droplet.

The space between the $+1$ and $-1$ hedgehogs is filled with a twisted LC, having the characteristic shape of an elongated cholesteric bubble, anchored with one end to the $+1$ defect, as shown in Fig. 1f. The cholesteric bubbles can be easily recognized in a streamline representation of the director field in Fig. 1g which shows only the projection of the director field on a plane. The white areas where the director is mostly perpendicular to the cross-section indicate the locations of the cholesteric bubbles. The bubbles are non-singular, axially symmetric, smoothly winding structures, which induce elastic repulsion between the attracting pairs of topological charges and stabilize the defect constellations. The cross-section of a cholesteric bubble is a two-dimensional skyrmion[29] of the Bloch type[35], as shown in Fig. 1h.

At a higher chirality parameter $N > 3$, we observe in different droplets many diverse constellations for the same value of $N$,

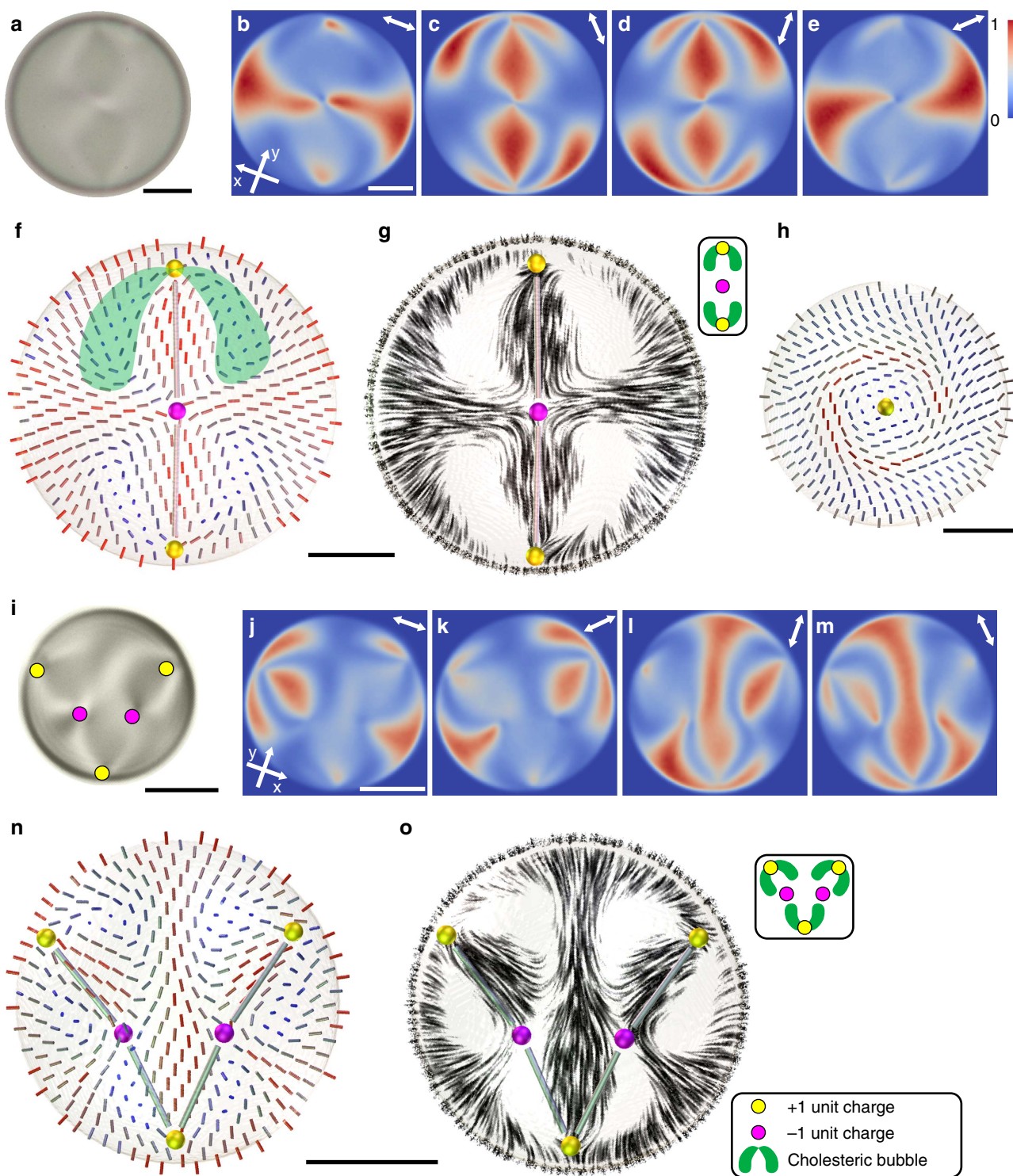

**Figure 1 | Unit topological charges separated by cholesteric bubbles in spherical chiral nematic droplets.** (**a**) A non-polarized micrograph of a droplet with $N = 3$. (**b**–**e**) Bleaching corrected, deconvolved and normalized FCPM intensities in an equatorial $xy$ plane of the droplet in **a**, for each of the four excitation/detection polarizations. Polarization for each panel is indicated with an arrow in top right corner. (**f**) Reconstructed director cross-section from the FCPM images showing a string of charge-alternating hedgehogs. The cylinders representing the director field are coloured by the director projection to the cross-section plane. The green-shaded area represents a cholesteric bubble. The thin rods that connect point defects indicate the spatial relation between the defects. (**g**) Director field from **f** in streamlines. Areas where the director is perpendicular to the cross-section have no streamlines, indicating the location of the cholesteric bubble. The schematic representation of the structure in the inset shows the relative positions of the point defects and the cholesteric bubbles. (**h**) Cross-section of director field perpendicular to the 3-point string in the middle of a cholesteric bubble. (**i**) A non-polarized micrograph of a $N = 3.1$ droplet with five unit charge hedgehogs, forming a V-shaped planar constellation in the focal plane. (**j**–**m**) Bleaching corrected, deconvolved and normalized FCPM intensities in an equatorial $xy$ plane of the droplet in **i**, for each of the four excitation/detection polarizations. (**n,o**) Director cross-section showing a V-shaped string of five unit charge hedgehogs shown in **n** cylinders and **o** streamlines. The inset to **o** shows a schematic representation of the structure in **i**–**o**. All scale bars are 5 μm.

which can be classified into two species: (i) strings of alternating point defects with unit hedgehog charges, (ii) constellations with a mixture of unit and multiple (that is, higher) hedgehog charges. In all cases, the $+1$ hedgehogs are found next to the surface, each anchored to its corresponding cholesteric bubble, while the negatively charged hedgehogs arrange in the bulk between the bubbles.

**String-like constellations**. Strings of alternating unit-charge hedgehogs are presented in Fig. 1i–o for 5 and in Fig. 2a,b and Fig. 2c,d for 7 and 9 unit-charge hedgehogs, respectively. Figure 1i shows a non-polarized micrograph of a droplet with a chirality parameter $N = 3.1$ and Fig. 1j–m show the corresponding experimental FCPM data. The reconstructed director field in Fig. 1n reveals a V-shaped constellation of five unit, charge-alternating, point defects and the streamline representation in Fig. 1o indicates the positions of three cholesteric bubbles between the point defects as is schematically shown in the inset. The three hedgehogs close to the surface are of the $+1$ radial type, whereas the two hedgehogs in the midst of both arms of the V-shaped constellation are the $-1$ hyperbolic type, so the total topological charge is $+1$. Similarly as in the droplet with three point defects (Fig. 1a–h), the cholesteric bubbles, anchored to the $+1$ hedgehogs, face the hyperbolic $-1$ hedgehogs between them.

Even longer strings with 7 and 9 unit charge hedgehogs are found at higher chirality (Fig. 2, Supplementary Movies 1 and 2), always containing an odd number of hedgehogs, with each added pair of $+1$ and $-1$ hedgehogs forming another bubble and extending the string of hedgehogs with a new arm as shown in the schematic representations in insets to Fig. 2b,d. Whereas the 5 unit-charge hedgehogs in Fig. 1i–o can be neatly packed into the equatorial plane of the droplet, the constellations with 7 and 9 unit hedgehogs in Fig. 2b,d pack in string-like structures in 3D.

**Higher topological charges**. Some hidden states are not formed only of unit topological charge hedgehogs, but also contain double ($q = -2$) and triple ($q = -3$) hedgehog charges as the binding elements. A double hedgehog charge appears in a planar configuration in Fig. 3a–c and Supplementary Movie 3 with a threefold optical symmetry containing three $+1$ hedgehogs close to the surface of the droplet and the fourth hedgehog (cyan dot) located exactly in the centre of the droplet. The three near-surface $+1$ hedgehogs together with the central hedgehog must add up to the $+1$ total topological charge of the droplet, which means that the central hedgehog has a $-2$ topological charge. Its threefold symmetry of binding to the surface unit hedgehogs (Fig. 3b) resembles a trivalent atom with three symmetrically positioned orbitals (for example, a carbon atom with $sp^2$ hybridized orbitals). This structure does not match any of the previously experimentally observed point defects in 3D nematics.

An even larger topological charge, carrying a hedgehog charge $q = -3$, is shown in Fig. 3d,e and Supplementary Movie 4. Here four unit-charge $q = +1$ hedgehogs form a tetrahedral constellation around the tetravalent topological charge in the centre of the droplet (green dot). Its topological charge $q = -3$ follows from the conservation of the overall topological charge. The $q = -3$ hedgehog forms a tightly squeezed director structure at the centre of the droplet, surrounded by four neighbouring cholesteric bubbles, which are compressing it towards the centre. The structure of this $q = -3$ hedgehog defect is fundamentally three-dimensional, reminiscent of a $sp^3$ hybridized atomic orbital.

Like in molecular chemistry, a single charged unit can be replaced by a compound structure with the same charge, while not changing the valence of the configuration. For example, in Fig. 4a,b and Supplementary Movie 5, one of the $+1$ hedgehogs from Fig. 3c is replaced by a string-like constellation of three hedgehogs, similar to the three-hedgehog string in Fig. 1a–g and likewise carrying a $+1$ topological charge. The tetravalent configuration from Fig. 3d,e also has a less symmetric variant, shown in Fig. 4c,d and Supplementary Movie 6 with six $+1$ hedgehogs in the vertices of an octahedron. Three of them are a part of the tetravalent configuration with a $q = -3$ point defect at the centre. The other three, together with two $q = -1$ hedgehogs,

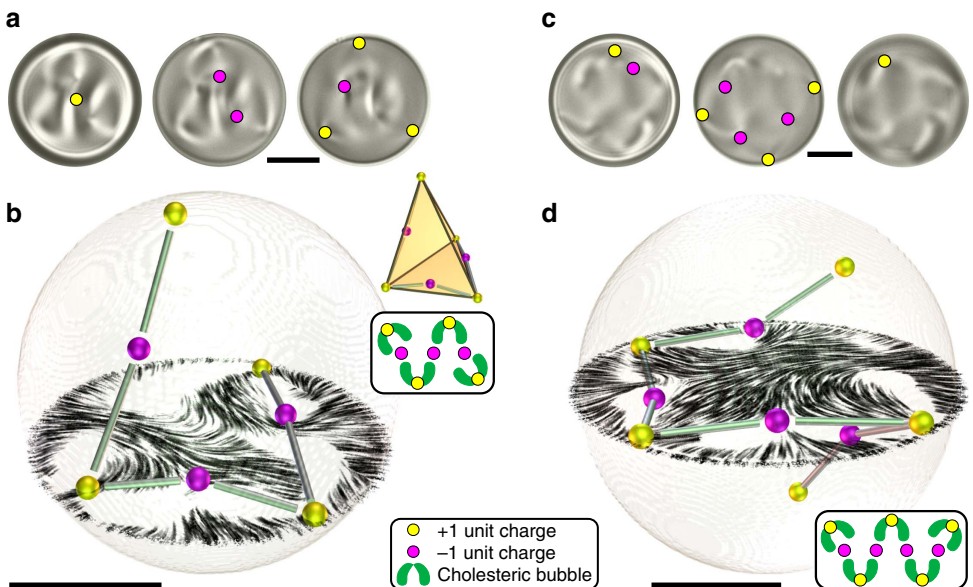

**Figure 2 | String-like constellations of unit topological charges in spherical chiral nematic droplets.** (**a**) Transmission images of a droplet ($N = 5.2$) with seven unit charge hedgehogs taken at different focusing depths. (**b**) Three-dimensional constellation of seven unit charge hedgehogs and a cross-section of the director field. Tetrahedral symmetry of the constellation is highlighted by the geometrical representation in the inset. Experimental FCPM intensities and complete reconstructed director are shown in Supplementary Movie 1. (**c**) Transmission micrographs of a droplet with a nine unit charge hedgehog string constellation. (**d**) Three-dimensional constellation of nine unit charge-alternating hedgehogs and a cross-section of the director field through the equator. Experimental FCPM intensities and complete reconstructed director are shown in Supplementary Movie 2. All scale bars are 5 μm.

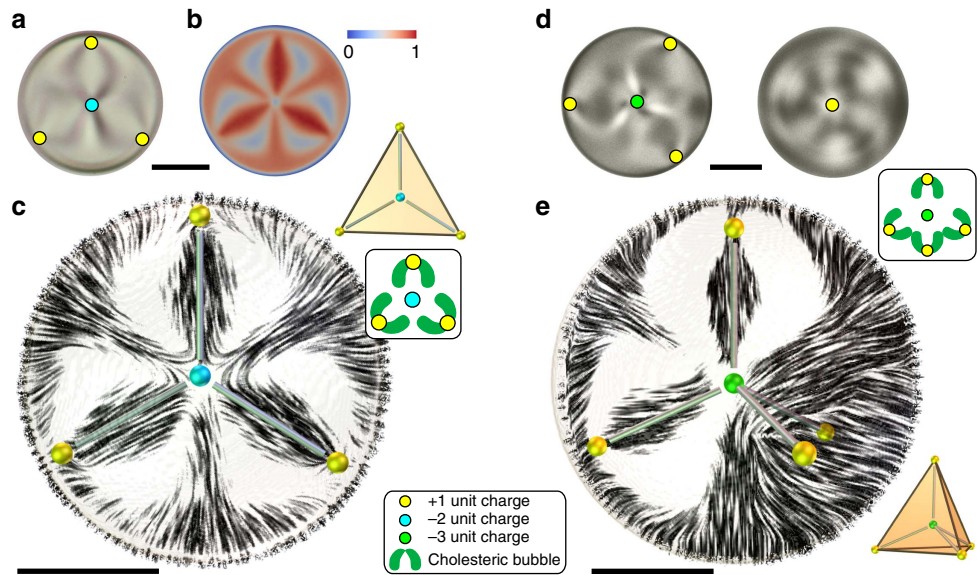

**Figure 3 | Higher topological charges in triangular and tetrahedral constellations.** (**a**) A micrograph of a droplet with three $q = +1$ point defects (yellow) and a higher charge $q = -2$ point defect (cyan). (**b**) FCPM total intensity $I_{tot}$ in the equatorial cross-section of the droplet in **a**. (**c**) Reconstructed director field, shown in streamline projection to the equatorial plane. The thin rods connecting the point defects show the spatial relation of the defects which is highlighted in the geometrical representation in the inset. The other inset schematically shows the point defects and the cholesteric bubbles. Experimental FCPM intensities and complete reconstructed director are shown in Supplementary Movie 3. (**d**) Micrographs of a droplet with a tetrahedral configuration of $q = +1$ hedgehogs and a triple charge $q = -3$ in the centre at different focuses, showing the positions of defects. (**e**) A 3D representation of the structure in a droplet with the $-3$ defect illustrating the tetrahedral configuration of $q = +1$ hedgehogs, which is highlighted in the inset where the $+1$ defects are shown in vertices of a tetrahedron. The streamlines show the director field in a plane which includes two $+1$ and the $-3$ defect, intersecting two cholesteric bubbles. Experimental FCPM intensities and complete reconstructed director are shown in Supplementary Movie 4. All scale bars are 5 μm.

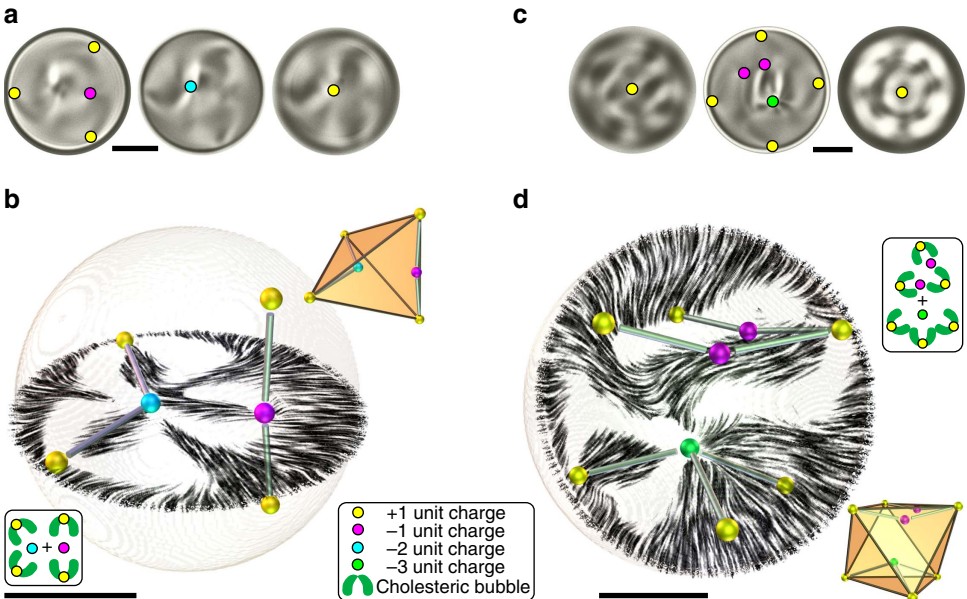

**Figure 4 | Tetrahedral and octahedral topological molecules with higher charge defects.** (**a**) Micrographs at different focusing depths showing the locations of point defects and (**b**) a 3D representation of the structure in a droplet similar to Fig. 3c, where one of the unit charges is replaced by a hedgehog molecule as can be seen in the schematic representation of the structure in bottom left inset. The streamlines show the reconstructed director in a plane which includes two $+1$, a $-1$ and the $-2$ defect. The thin rods connecting the point defects show the spatial relation of the defects, which is highlighted by the geometrical representation in the top right inset where the $+1$ defects are shown in the vertices of a tetrahedron. Experimental FCPM intensities and complete reconstructed director are shown in Supplementary Movie 5. (**c**) Non-polarized micrographs and (**d**) a 3D representation of a structure in which one of the $+1$ charges in the tetrahedral constellation from Fig. 3e is replaced by a V-shaped string of five unit charges as shown in the schematic inset. By this substitution the symmetry of the structure changes to octahedral, as illustrated in the lower right inset. Experimental FCPM intensities and complete reconstructed director are shown in Supplementary Movie 6. All scale bars are 5 μm.

form a V-shaped string with a total charge $q = +1$, which acts as the fourth $+1$ hedgehog of the tetrahedral configuration. The stability of these structures demonstrates that the high-charge defects do not occur strictly in a confinement field with just the right symmetry, but can also withstand exchanges of the surrounding hedgehogs with larger constellations.

## Discussion

The experiments shown in Figs 3 and 4 and Supplementary Movies 3–6 constitute the first-ever observation of higher point topological charges in 3D. In NLCs, the topological charge is measured by counting the number of patches of the director field that are piercing the sphere which is enveloping the defect[36]. Figure 5a,b show these patches, formed by the reconstructed director field around the $-2$ defect, which is decorated continuously with arrows; a patch is a region where the director field points outwards (red regions) on a continuous background of inward-pointing director (blue regions; note that this choice of arrows is opposite to the one we chose for the calculation of topological charge and its sign to make patches visually more discernible). Each patch can be understood as a bundle of hyperbolic director streamlines in Fig. 5c that converge towards a neighbouring $+1$ hedgehog. A defect with $M$ patches has a hedgehog charge of $q = 1 - M$ (ref. 36); therefore, the defect

in Fig. 5a,b,d,e has a $q = -2$ charge and Fig. 5f shows a $q = -3$ charge. The patch structure reveals that $q = -2$ and $q = -3$ defects are generalizations of the hyperbolic hedgehog $q = -1$, and form at the confluence of streamlines from the $+1$ charges below the droplet surface, similar to how saddle points occur in an electric field formed by a set of equally charged point charges.

Highly symmetric constellations are obtained for the higher charges, which reside in the centre of the droplet: the $q = -2$ charge stabilizes the triangular molecule in Fig. 5g, the $q = -3$ charge binds the tetrahedral molecule in Fig. 5h, and the $q = -5$ the octahedral one in a model in Fig. 5i. The valence properties of the higher topological charges in chiral nematic spheres are similar to the tetrahedral structure of the monovalent topological defects in nematic shells[37] which were proposed to be used for realization of tetravalent colloids[38].

It is quite interesting that we find the $+1$ hedgehogs close to the surface in all our experiments on topological hidden states in cholesteric droplets. This can be understood by considering the growth of the cholesteric regions (the cholesteric bubbles) after the quench. It is energetically favourable to expel the $+1$ hedgehogs towards the surface, where the helical order cannot exist because of the $\mathbf{n} \cdot (\nabla \times \mathbf{n}) = 0$ surface constraint. The homeotropic surface itself naturally prefers a general radial-like structure of the near-surface defects, which is why the defects at

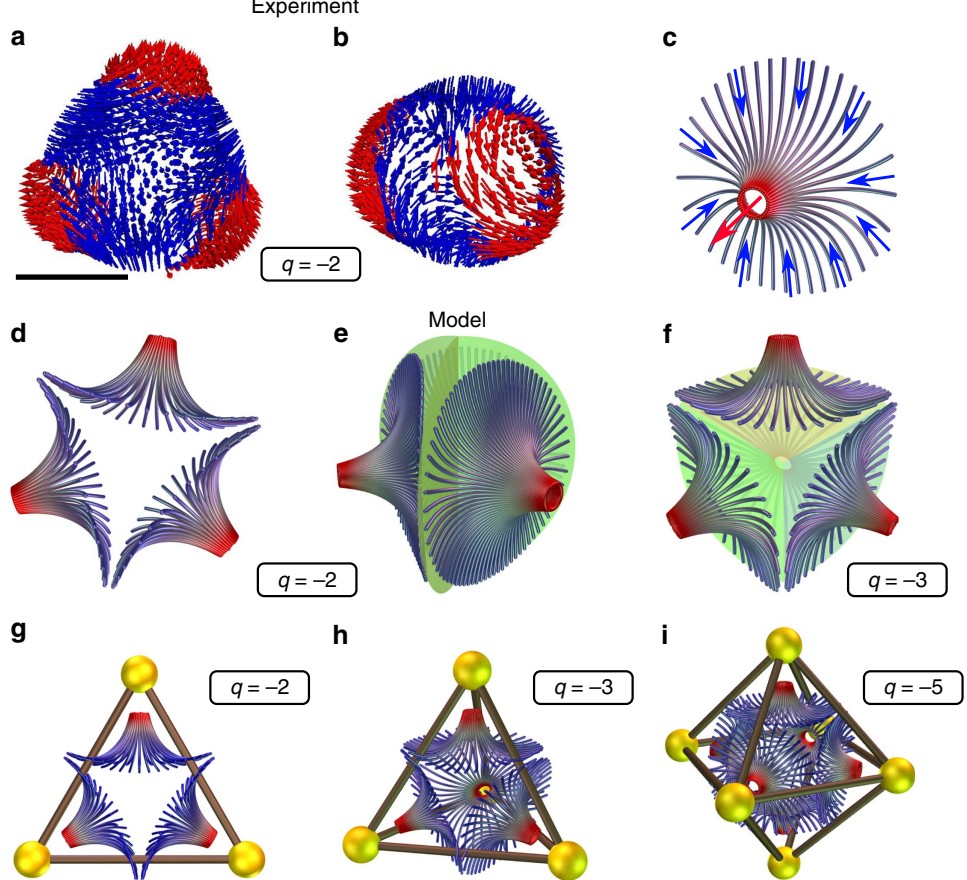

**Figure 5 | Topological valence chemistry with higher topological charges.** (**a,b**) Patch structure from the FCPM experimental data for a $-2$ defect, from two perspectives. Notice the outgoing arrows at three valence directions. The direction of arrows on **a–c** is for the sake of visual clarity of this figure inverted with respect to the directions we used for calculation of topological charge. (**c**) A single patch structure represented as streamlines as an attachment point for a $-1$ hedgehog. (**d,e**) A sketch of director streamlines for $-2$ defect, showing three patches. (**f**) A sketch of director streamlines for a tetrahedrally symmetric $-3$ defect; the fourth patch is pointing away from the reader. (**g–i**) Topological defect constellations with polyvalent charges. The $+1$ defects near the droplet surface are bound by the polyvalent negative charge in the centre of the droplet. See Fig. 3c,e for realizations of **g,h**, respectively. The scale bar is 2 μm.

the surface all have the same form and the same unit charge $+1$. Defects with positive topological charge greater than $+1$ cannot form close to the interface—they are not equivalent to hyperbolic saddle points, which occur naturally between other defects.

In quasi-two-dimensional systems, the higher topological charges are, in most cases, unstable, as an equivalent number of smaller defects or disclinations achieves a lower free energy due to a lower elastic distortion, but were observed in specific experiments[39–41]. It was shown theoretically that in 3D achiral nematics the hedgehogs with higher charges than $\pm 1$ are unstable[42]. Here they are stabilized by chirality (Fig. 6), which acts as a stabilizing spring inside the spherical confinement, preventing dissociation of the higher-charge defects. As we can see from the stability chart Fig. 6, several different structures can appear at a given relative chirality $N$. This implies that the state which will emerge in a droplet after the quench is not determined by the helical pitch and droplet size; instead, the metastable states appear with probabilities which depend on their relative energies[17,33]. The most complex hidden states appear only in the middle of the stability range of structures with the same number of positive charges (Fig. 6). This shows there is an optimal ratio between chirality and confinement, which promotes the formation of higher charge topological defects.

The higher charge defects could in general be true point defects, or have the internal structure of a defect loop, analogous to recently demonstrated defect loops in the cores of $+1$ defects[43], but with a more complex structure of its cross-section[44]. Alternatively, higher charge defects could be composed of several tightly packed topological point defects, for example, the $q = -2$ could be composed of three $-1$ defects and one $+1$ defect. However, the FCPM image in Fig. 3b shows that the $q = -2$ defect is confined to a volume comparable to a $-1$ defect. Any internal structure would have to be confined to a volume smaller than the resolution of our microscope, which is $\sim 300$ nm. This is an order of magnitude smaller than the helical period of the LC mixture, which sets the minimum separation between the defects for which the helical twist still acts as a repulsive force[30]. We can, therefore, safely conclude that the central topological defect is a single point defect, carrying the topological charge $q = -2$.

Our experiments in chiral nematic droplets clearly reveal, for the first time, the full anatomy of the hedgehog defects, as we are able to reconstruct the topology of the 3D director and perform a patch analysis of the streamlines. Higher topological charges,

which were previously considered unstable are in fact strongly stabilized by the chirality and spherical confinement of the NLC droplets on a time scale of several days. Surprisingly, the topological defects can be arranged in a perfect polyhedral constellation inside a liquid sphere, which has implications far beyond the field of LCs. Our strategy, which combines spherical confinement and a chiral field, offers guidance for the formation of hidden skyrmionic constellations in chiral magnets, cold atoms, and polyvalent soft-matter colloidal science. Regular packing of the topological defects into a sphere can be regarded as the synthesis of topological molecules, formed of polyvalent defects, where the skyrmion-like structures have the role of topological bonds.

From a practical perspective, chiral nematic droplets with higher topological charges in the centre could serve as polyvalent colloidal particles, providing directional bonds to the surrounding colloidal particles. For example, a $q = -3$ topological charge in the centre of the chiral nematic droplet provides a tetravalent coordination of unit charge $q = +1$ hedgehogs located close to the surface of the droplet. If the surroundings of these $q = +1$ hedgehogs are decorated with polymer linkers, similar to the idea of Nelson for the nematic shells[38], a cholesteric droplet would be similar to a 4-valent atom like carbon, silicon and germanium. Here the polymer linkers could provide artificial bonds to neighbouring droplets for self-assembly into colloidal crystals with a diamond lattice.

## Methods

**Preparation of droplets.** The LC droplets were prepared with a low-birefringence LC mixture of 1:1 weight ratio of 4′-butyl-4-heptyl-bicyclohexyl-4-carbonitrile (CCN-47) and 4,4′-dipentyl-bicyclohexyl-4-carbonitrile (CCN-55, both purchased from Nematel; refractive indices of the mixture $n_o = 1.47$ and $n_e = 1.50$) doped with 1–2% of chiral dopant S-811 (Merck) to get a mixture with a pitch in the range $\sim 6$–$12\,\mu m$. A small amount of dye N,N′-bis(2,5-di-tert-butylphenyl)-3,4,9,10-perylenedicarboximide (BTBP, Sigma Aldrich) was added to the LC mixture to enable FCPM by first dissolving the dye in acetone, adding the dye/acetone solution to the LC mixture and evaporating the solvent at room temperature to achieve a homogeneous mixture. Droplets with diameters in the range $\sim 10$–$20\,\mu m$ were produced by mixing the LC mixture in a glycerol medium ($n_{glycerol} = 1.47$) doped with 4% wt. L-α-phosphatidylcholine (lecithin, Avanti Polar Lipids) to achieve perpendicular orientation of LC molecules on the droplet surface. The medium with the droplets was sandwiched between a microscopic cover glass of $150\,\mu m$ thickness and a thicker 1 mm glass, which were separated by $30\,\mu m$ mylar spacers. The cell was sealed along the perimeter with a fast curing two-component epoxy glue to prevent droplet movements because of glycerol flow. Constellations of point defects formed after the sample was quenched from the isotropic phase at a rate of several $K\,s^{-1}$ to room temperature.

**FCPM microscopy and director reconstruction.** The structures in the droplets were imaged by fluorescence confocal polarizing microscopy (FCPM)[45] on a Leica TCS SP5 X confocal microscope with a Leica WLL laser light source. The microscope was modified by inserting a quarter-wave plate for the selected excitation wavelength (488 nm) in the slot below the objective to transform the polarization of the microscope from linear to circular. Linear polarizers were inserted above the waveplate into the same slot as the waveplate to select different linear polarizations of excitation. A full $xyz$-scan was preformed at four linear polarizations in the $xy$ plane, separated by 45° to obtain intensities $I_0$, $I_{\pi/4}$, $I_{\pi/2}$ and $I_{3\pi/4}$. The scan at the first polarization was repeated in the end, to be used for linear correction of intensities because of bleaching between the scans. The $xyz$-stacks were then deconvolved with SVI Huygens Professional software. Orientation of the director in the $xy$ plane was calculated from the intensities as[21]:

$$\phi = \frac{1}{2}\arctan\frac{I_{\pi/4} - I_{3\pi/4}}{I_0 - I_{\pi/2}}. \tag{1}$$

Total intensity $I_{exp}$ was calculated as: $I_{exp} = \frac{1}{2}(I_0 + I_{\pi/4} + I_{\pi/2} + I_{3\pi/4})$ and $z$-dependent background $I_{offset}$ and normalization $I_{norm}$ corrections were applied to calculate the out-of-plane angle $\theta$ from equation[30]: $I_{exp} = I_{offset} + I_{norm}\cos^4\theta$. The director field was calculated from $\theta$ and $\phi$ as: $\mathbf{n} = (\cos\theta\cos\phi, \cos\theta\sin\phi, \pm\sin\theta)$. A simulated annealing algorithm[30] was used to determine the sign of the $z$ component of the director field by randomly selecting a point in the droplet and calculating the elastic energy of director deformation in that point from:

$$f_e = \frac{1}{2}L\left(\frac{\partial Q_{ij}}{\partial x_k}\frac{\partial Q_{ij}}{\partial x_k} + 4q\epsilon_{ikl}Q_{ij}\frac{\partial Q_{lj}}{\partial x_k}\right), \tag{2}$$

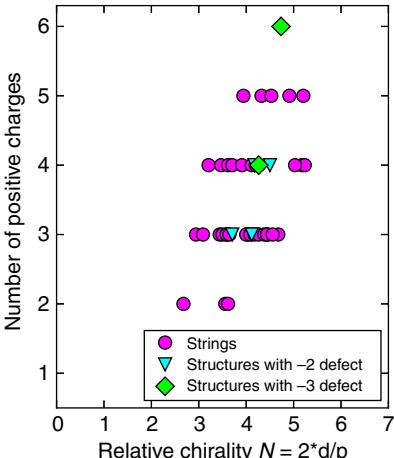

**Figure 6 | Stability chart of topological defect strings and polyhedral constellations with higher charges.** Note that for each number of defects, the higher hedgehogs are concentrated in the middle of the chirality range.

where $L$ is the elastic constant in the one constant approximation and $q = 2\pi/p_0$ is the inverse cholesteric pitch, which makes the elastic energy chirality-dependent. The tensor $Q$ is calculated from the director as

$$Q_{ij} = \frac{S}{2}\left(3n_i n_j - \delta_{ij}\right), \tag{3}$$

where the scalar order parameter $S$, which is not available from the FCPM experiment, is taken to be constant. This elastic energy is compared with the elastic energy in that point with flipped $z$-component of director field $(n_x, n_y, -n_z)$. If the energy of the flipped state is lower than the starting energy, this state is kept, but if it is higher, the flip is accepted with a probability given by a Boltzmann weight $\exp(-\Delta E/t)$, where $\Delta E$ is the energy difference between the two states and $t$ is a free parameter. The algorithm starts at a large $t$, iterating the flipping procedure over the whole volume of the droplet until a stable energy of the whole structure is reached, and then $t$ is lowered and the procedure is repeated until thermalization of the structure is achieved. This procedure scans all possible configurations of signs of $n_z$ and finds the ones with the lowest elastic energy within the restraint of experimental data for $\theta$ and $\phi$, which means it is the best approximation of the observed structure that can be found from experimental data[30].

To generate the streamline representations of the director, the director field in a cross-section of the sample was projected on the cross-section plane and smoothed to indicate the average local direction. Areas where the projection of the normalized director is smaller than 1/3 (where director is mostly perpendicular to the cross-section) are excluded from the final streamline presentation.

**Data availability.** All the data relevant to the findings of this study are available from the authors upon request.

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

## Acknowledgements

G.P. acknowledges the financial support of Slovenian Research Agency (ARRS) through contract PR-05014. S.Č. acknowledges the financial support of ARRS through contracts Z1-6725 and P1-0099. I.M. acknowledges the financial support of ARRS through contracts J1-6723 and P1-0099.

## Author contributions

G.P. conducted the experiments, G.P. and S.Č. analysed the results, I.M. conceived and supervised the experiments and wrote the main manuscript. All authors contributed to the final version of the manuscript.

## Additional information

**Publisher's note**: 

