## [Peer Review File · Nature Communications]

Reviewers' comments:

Reviewer #1 (Remarks to the Author):

This paper has presented long-lived hidden topological defects in chiral nematic droplet by varying chirality, N , showing string-like, triangular and polyhedral constellations of monovalent and polyvalent singular point defects. In this work, indeed, for the first time, the 3-dimensional construction of topological point defects with hedgehog charge $q=-2, -3$, has been concretely presented based on the FCPM images. In terms of the inside structure of the chiral nematic droplet, this work is quite interesting and important in topological defects of the liquid crystal. However, for the clarity, there are some parts to be revised. For example, the reconstructed director field in figures 1 and 2 based on the FCPM image is not quite easy to understand because given FCPM images are not clear (e.g., two pink points positioned in the middle of droplet in figures 1f,g are not clear to see). Moreover, this manuscript does not give the discussion for the considerable factors to fabricate droplets that show the multi-unit charge hedgehog. After the revision, I will recommend this paper for the publication in Nature communications.

Q1. In this paper, all the data were constructed based on FCPM images, however, there is much unclarity in the corresponding FCPM images (figures 1 and 2) that are too much overlapped by the sketch of director field to see how they look like. I recommend the figures including optical micrograph, FCPM images, and corresponding reconstructed sketches should be well-reorganized for the clearness.

Q2. It is not clear how to reconstruct the director field or director streamlines in figures 1 and 2 from FCPM images. This explanation will be helpful to understand the situation during the reconstruction.

Q3. There is a size distribution in the diameter of chiral nematic droplets, 10 – 20 μm . Since the chirality is determined by diameter of droplet and intrinsic pitch of chiral nematic droplet, effect of size distribution on the formation of topological defects in the droplet needs to be discussed.

Q4. In line 82 and 83 of the page 6, author said 'It does not match any of the previously experimentally observed point defects in 3D nematics' for the figure 2a, but a paper published in Nat. Comm. 7603(2016), doi:10.1038/ncomms8603 has already shown the similar image in figure 1i, the three-fold symmetry connecting to the surface hedgehogs. You should discuss this comparing yours and the previous result to insist the novelty.

Q5. In figures 2 and 3, there are many states of topological structures in the droplets as shown in the optical images. I'd like to ask questions for this. Are they degenerated? Or is there more frequently emerged structure among the states? If yes, can you explain the reason why one of states is more frequently emerged? What makes it stable compared with other states? These things should be fully discussed in the revised manuscript.

Q6. It is mentioned in 116-118 lines of page 8 and 131-133 lines of page 9, "Higher topological charges, which were previously considered unstable are in fact strongly stabilized by the chirality and spherical confinement of the NLC droplets on a time scale of several days", meaning that they are under the equilibrium state. If so, are they reversible during the transition from isotropic to nematic phase? I think the stability issue considering temperature variation is important in its application or control.

Q7. Once again this is very interesting in science, but could you give some perspective views in some applications to convince the readers they think your work is important?

Reviewer #2 (Remarks to the Author):

In the manuscript "Hidden topological constellations and polyvalent charges in chiral nematic droplets" the authors have discovered the existence of 3D point defects of large negative topological charges in chiral nematic droplets quenched from the isotropic phase. The successful observation became possible due to adjusting the ratio between the droplet diameter and the cholesteric pitch, and due to employing their previously developed method to decipher the nematic orientational field from multiple Fluorescence Confocal Polarizing Microscopy images [18]. The large negative defects are surrounded by other defects because the total topological charge of the defects in a droplet should be +1, creating the topological analog of planar, tetrahedral and octahedral molecules. The obtained results will be of interest for a broad readership of physicists and chemists.

In my opinion the manuscript deserves to be published in Nature Communications; however, the discussion why only negative large topological defects were observed would improve the text.

Sergij V. Shiyanovskii
Kent State University.

Reviewer #3

This is a wonderful paper on point defects in small cholesteric liquid crystal drops. The authors show that such systems exhibit organised, often highly symmetric, configurations of point defects, including those with topological charge -2 and -3 , which hitherto have not been seen in any material. This is clearly of fundamental importance in understanding how the topological properties of materials can be harnessed and controlled. The primary significance is for liquid crystals but, given that there are only point defects, the same phenomena should arise also in chiral ferromagnets, which are governed by essentially the same energy functional. Thus I believe there should be both widespread interest and cross-disciplinary implications. The paper is well-written and clear enough for me to understand and follow the main concepts, even in the short time-scale of the review process. I have no hesitation in recommending that it be published in Nature Communications.

I have only a small number of minor comments and suggestions that should be treated entirely at the authors' discretion.

It is perhaps slightly unusual to refer to the Gauss-Bonnet theorem (line 44) for the constraint on the total charge enclosed by a surface. It is fine, but personally I think more in terms of homotopy invariance of degree and Hopf's index theorem for vector fields.

In line 56, the sentence "*At a higher chirality parameter $N > 3$, for the same value of N , we observe ...*" reads to me to be self-contradictory and I think the construction "*At a higher chirality parameter $N > 3$, we observe many diverse constellations, for the same value of N , ...*" works better, without altering what I believe you mean to say in this sentence.

The paragraph from line 84 to line 87 describing the charge -3 defect is quite short, especially given the emphasis that is placed upon this result throughout your paper. It is true that the following paragraph gives some elaboration and there is no need to not be so succinct, but it is perhaps worth thinking about how brief the statement of this central result is.

I would like you to verify the direction assignment for arrows in your Figure 3. You write on line 48-49 that the orientation is outward from the droplet surface. On this basis I would assign precisely the opposite direction as you have done in Figure 3, so I would ask you to check to make sure you are correct.

In line 126, I would advocate a moderately more reserved statement by removing the word “possible” from “... *minimum possible separation* ...”.

In lines 129-130 you write “*Our experiments in chiral nematic droplets clearly reveal, for the first time, the full anatomy of the hedgehog defects ...*”. I think they do and I think this is an excellent achievement, and I think it is possible, by staring intently at the figures you have provided, to get a feeling for the structure of higher charge point defects, but it is not very easy. Part of this is to get a feeling for the nature of the chirality in these droplets. Minimisation of the elastic energy will require that the director preferentially attains a configuration in which $\mathbf{n} \cdot \nabla \times \mathbf{n} \approx -q_0$ locally. The extent to which this is achieved (or, conversely, frustrated), and how it is, is not clear, to me at least, although I do appreciate that this is indicated by your ‘cholesteric bubbles’. Nor is it immediately clear where the defects in the pitch axis (λ lines) are, or what their structure is like. It would be fascinating to figure all this out. I regret that I do not have anything useful to suggest to you, but still thought I would encourage you to continue thinking about such structures and promote your contribution.

As a final comment I might say that one of the striking features of your experiments is that +1 defects are attracted to the surface of the droplet, while negative charge defects are pushed to the interior. In a nematic droplet, the simplest configuration one can think of has a single +1 defect at the centre; it is not attracted to the surface. One may reasonably ask why the cholesteric droplets do not simply produce a chiral distortion of this configuration, especially when the pitch is comparable to the droplet size. Accounting for this contrast seems to me to be a first challenge for theory.

Reviewer #1:

This paper has presented long-lived hidden topological defects in chiral nematic droplet by varying chirality, N , showing string-like, triangular and polyhedral constellations of monovalent and polyvalent singular point defects. In this work, indeed, for the first time, the 3-dimensional construction of topological point defects with hedgehog charge $q=-2, -3$, has been concretely presented based on the FCPM images. In terms of the inside structure of the chiral nematic droplet, this work is quite interesting and important in topological defects of the liquid crystal. However, for the clarity, there are some parts to be revised. For example, the reconstructed director field in figures 1 and 2 based on the FCPM image is not quite easy to understand because given FCPM images are not clear (e.g., two pink points positioned in the middle of droplet in figures 1f,g are not clear to see). Moreover, this manuscript does not give the discussion for the considerable factors to fabricate droplets that show the multi-unit charge hedgehog. After the revision, I will recommend this paper for the publication in Nature communications.

Reply: We thank the reviewer for all the comments. We have carefully considered all of them and prepared a point-by-point answers to address them. We also took into account your general remarks when reorganizing the manuscript to Nature Communications format. We trust you will find the amended text of better quality.

Q1. In this paper, all the data were constructed based on FCPM images, however, there is much unclarity in the corresponding FCPM images (figures 1 and 2) that are too much overlapped by the sketch of director field to see how they look like. I recommend the figures including optical micrograph, FCPM images, and corresponding reconstructed sketches should be well-reorganized for the clarity.

Reply: Thank you very much for this very important comment on the presentation of data. We have been struggling a lot to present all the relevant data from our FCPM images in an understandable way. This is not easy, because one needs to present the essential physics captured by 3D FCPM imaging. For

the purpose of clarity of visual presentation of our results, we made the following changes and amendments in our revised manuscript:

- We have taken the reviewer's comments about overlapping of the images into account and replaced the polyhedral solids in the images with schematic ones in insets to stress the spatial symmetry of the constellations. Figures 1 and 2 are now new.
- We added to the Supplementary material experimental FCPM intensities for two droplets in which the defects lie mostly in a single plane together with the reconstructed director field and the streamlines generated from it. We trust that these intensities and the description of the director reconstruction procedure give the reader sufficient information in a clear way. The non-planar structures on the other hand are much more difficult to present with intensity cross-sections and we believe that the projections with positions of the point defects are the best option for a publication.
- In addition to this, we prepared 6 new composite movies of the 3D FCPM scans together with the corresponding cross-sections of 3D reconstructed director fields for all the presented structures and included them as Supplementary material. The movies now present in parallel the FCPM intensities at a given cross section (for different polarizations) and the reconstructed director field. We trust this is the best way to present our 3D data, as the reader can choose any cross section and compare the FCPM scans and director reconstruction.

We should also mention that the experiments and reconstruction protocols were also explained in our first paper on this subject, published in *Scientific Reports* **6**, 26361 (2016), reference #30 in present manuscript.

Q2. It is not clear how to reconstruct the director field or director streamlines in figures 1 and 2 from FCPM images. This explanation will be helpful to understand the situation during the reconstruction.

Reply: Thank you for this comment. The reconstruction procedure was originally described in the Supplementary material, but it is now included in the main text in the Methods section, following the formatting standard of Nature Communications. We hope the description of the method together with additional FCPM data in the supplement is now sufficient for the reader to understand how the director fields are reconstructed from the experimental data and repeat the procedure. Additional information is available in our first paper on the reconstruction, published in *Scientific Reports* **6**, 26361 (2016), reference #30 in this manuscript.

Q3. There is a size distribution in the diameter of chiral nematic droplets, 10 – 20 μm . Since the chirality is determined by diameter of droplet and intrinsic pitch of chiral nematic droplet, effect of size distribution on the formation of topological defects in the droplet needs to be discussed.

Reply: Thank you for this comment. The size distribution is indeed crucial if one wants to make any claims on probability of the appearance of different topological structures in the droplets. However, the pitch and the droplet size are not the only parameter influencing this probability. We find that the rate of cooling during the quench is the most important variable for obtaining constellations with multiple or higher charge point defects. There are several practical obstacles to studying the frequencies of appearance of the topological structures. For example, the dispersion of droplets in the measuring cell consists of thousands of droplets and it would be very difficult to characterize all of them. The imaging process for a single droplet typically takes around 15 minutes and additionally a couple of hours are needed to analyze the data for each droplet. This means that complete characterization of just one sample would be a massive undertaking. Even more likely, details in the quench procedure can strongly influence the probabilities of appearance of different structures so it is very difficult to compare results of different runs of the experiment.

Taking this into account, it made more sense to first search for novel structures that can be formed within this experiment and leave the characterization of frequencies of appearance of these structures for future work with differently designed experiments. Indeed the reconstruction of director fields in the droplets and its analysis poses a substantial task and requires ample text for presentation so we decided to omit any deeper excursions into relative frequencies of the structures from this article. What we did try to present are the chirality ranges at which the various structures can appear (Fig. 3j) which clearly show that the stability ranges of the structures considerably overlap indicating that the states are metastable and give the reader information on where to look for these structures in LC or other systems. This is now more emphasized and explained in the revised manuscript.

Q4. In line 82 and 83 of the page 6, author said 'It does not match any of the previously experimentally observed point defects in 3D nematics' for the figure 2a, but a paper published in Nat. Comm. 7603(2016), doi:10.1038/ncomms8603 has already shown the similar image in figure 1i, the three-fold symmetry connecting to the surface hedgehogs. You should discuss this comparing yours and the previous result to insist the novelty.

Reply: The image from the article of Orlova et al. Nature Communications 6:7603:DOI: 10.1038/ncomms8603 (2015) is indeed similar to Fig. 2a in our article. The authors of that article observe different topological states in chiral droplets but do not try to reconstruct the director fields in the droplets so the position and nature of topological defects could not conclusively be determined. It is not possible to make any firm conclusions based just on wide-field micrographs. However, from our experience and after analyzing very large number of droplet structures, we think the droplet in Figure 1e and 1l by Orlova et al. does not show three surface hedgehogs. The locations of surface hedgehogs can be more clearly identified in polarized images, indicated by two scythe-like bright lines converging to the surface point defect. For example, they can be clearly seen in Figs. 1c and 1d in the paper by Orlova et al. Fig. 1c is a crossed polarizers image of a droplet with one point defect close to the surface. Figure 1d from Orlova et al. shows a droplet with three colinear defects, which is the same structure as Figs. 1a-c in our article. In the polarized image in Fig. 1e of Orlova et al. we can see that only two of the supposed three point defects show these features from which we conjecture that there are only two surface

hedgehogs. In fact the polarized image Fig. 1e of Orlova et al. appears to be a slightly deformed version of Fig. 1d, which shows three colinear point defects. This leads us to conclusion that the structure in Fig. 1e of Orlova et al. is a variant of the droplet with three unit charge point defects, but in this case the hyperbolic point defect is not positioned in the center of the droplet but is moved in the top right direction of Figs. 1e,l. The dark lines which appear to hint at a position of a third surface point defect in Fig. 1l are therefore just a result of characteristic optical lensing from cholesteric layers and do not indicate any additional point defect.

Q5. In figures 2 and 3, there are many states of topological structures in the droplets as shown in the optical images. I'd like to ask questions for this. Are they degenerated? Or is there more frequently emerged structure among the states? If yes, can you explain the reason why one of states is more frequently emerged? What makes it stable compared with other states? These things should be fully discussed in the revised manuscript.

Reply: These are just a few of different structures that appear in chiral droplets at a given size and chirality and we are not yet in a position to characterize the frequency of their appearance. For a given relative chirality several of the presented structures can be observed implying that, even though they are all stable, most of them do not correspond to the ground state but are metastable states, separated between each other by high enough energy barriers to ensure stability. The stability chart of the structures is shown in Figure 3j. It is easy to see that several of the presented structures are stable at a given N . The presented structures are not the most frequent ones, but were chosen for this article based on their novelty and implications for the topology of materials. Other structures will be presented in future work where we will also address the probability of their appearance.

Q6. It is mentioned in 116-118 lines of page 8 and 131-133 lines of page 9, "Higher topological charges, which were previously considered unstable are in fact strongly stabilized by the chirality and spherical confinement of the NLC droplets on a time scale of several days", meaning that they are under the equilibrium state. If so, are they reversible during the transition from isotropic to nematic phase? I think the stability issue considering temperature variation is important in its application or control.

Reply: Thank you for raising this very important question, which is indeed crucial for control of the structures and their applications. The structures are stable in the sense that they do not spontaneously degrade over time (i.e. days). At a given chirality several structures are possible, indicating the structures are metastable. They must be separated by high enough energy barriers to prevent them from spontaneously undergoing structural transitions between them.

However, when a droplet with a given structure is heated into the isotropic phase and cooled down quickly, it does not return to the same structure and there is no memory effect. In the second quench the same droplet may appear as a completely different structure. So we could say there is a certain probability for a given structure to be formed during the quench of the very same droplet.

So far we observed the higher defects in samples in a specific quenching regime of a degree per second. Besides using specific cooling rates it might be possible to somewhat control the structures using laser beams of specific shapes.

Q7. Once again this is very interesting in science, but could you give some perspective views in some applications to convince the readers they think your work is important?

Reply: The higher topological defects could, because of their non-trivial valence, be used as building blocks in tailored self-assembled colloidal materials, e. g. tetravalent colloids. This is in line with the idea of Nelson on topological defects in nematic shells, Reference #38. We have now added an extra paragraph to explain these ideas for application in colloidal science.

Reviewer #2

In the manuscript “Hidden topological constellations and polyvalent charges in chiral nematic droplets” the authors have discovered the existence of 3D point defects of large negative topological charges in chiral nematic droplets quenched from the isotropic phase. The successful observation became possible due to adjusting the ratio between the droplet diameter and the cholesteric pitch, and due to employing their previously developed method to decipher the nematic orientational field from multiple Fluorescence Confocal Polarizing Microscopy images [18]. The large negative defects are surrounded by other defects because the total topological charge of the defects in a droplet should be +1, creating the topological analog of planar, tetrahedral and octahedral molecules. The obtained results will be of interest for a broad readership of physicists and chemists.

In my opinion the manuscript deserves to be published in Nature Communications; however, the discussion why only negative large topological defects were observed would improve the text.

Reply: We thank the reviewer for the favorable opinion and for recognizing the novelty of the work. We have carefully considered the suggestion on how to improve the article and have now expanded the text to discuss this interesting detail. All the observed structures have +1 hedgehogs at the droplet surface and negatively charged hedgehogs in the bulk. The droplet surface requires a radial-type geometry of the hedgehog, which are all assigned the +1 topological charge, because the director is pointing normal to the surface of the droplet. Higher topological charges of positive sign are disfavored both energetically and due to geometric constraints at the surface.

In the bulk, the positive topological charge has to be compensated, which happens in the form of saddle-points of the director field streamlines, as can be seen in Fig 3. These all have negative topological charge due to our initial choice of the direction, and are generalizations of the hyperbolic structure of a -1 hedgehog. Note that these structures are created after a temperature quench, and the cholesteric bubbles, interspaced by the negative defects, push against the positive hedgehogs, expelling them towards the surface. A cholesteric liquid crystal with well-defined pitch axis does not allow point

defects, and the anchoring condition requires $\mathbf{n} \times (\nabla \times \mathbf{n})=0$ at the surface, so the surface is a natural location for the defects for the cholesteric, whereas in a nematic, there is no such frustration.

Reviewer #3

This is a wonderful paper on point defects in small cholesteric liquid crystal drops. The authors show that such systems exhibit organized, often highly symmetric, configurations of point defects, including those with topological charge -2 and -3, which hitherto have not been seen in any material. This is clearly of fundamental importance in understanding how the topological properties of materials can be harnessed and controlled. The primary significance is for liquid crystals but, given that there are only point defects, the same phenomena should arise also in chiral ferromagnets, which are governed by essentially the same energy functional. Thus I believe there should be both widespread interest and cross-disciplinary implications. The paper is well-written and clear enough for me to understand and follow the main concepts, even in the short time-scale of the review process. I have no hesitation in recommending that it be published in Nature Communications.

I have only a small number of minor comments and suggestions that should be treated entirely at the authors' discretion.

Reply: We thank the referee for careful reading of our manuscript, positive response and all the comments. We have carefully considered them and amended the text where needed. We hope the referee will find our changes to improve the text.

Remark 1: It is perhaps slightly unusual to refer to the Gauss-Bonnet theorem (line 44) for the constraint on the total charge enclosed by a surface. It is fine, but personally I think more in terms of homotopy invariance of degree and Hopf's index theorem for vector fields.

Reply: Thank you for this suggestion, we now use the more common name - the Poincaré-Hopf theorem.

Remark 2: In line 56, the sentence "*At a higher chirality parameter $N > 3$, for the same value of N , we observe ...*" reads to me to be self-contradictory and I think the construction "*At a higher chirality parameter $N > 3$, we observe many diverse constellations, for the same value of N , ...*" works better, without altering what I believe you mean to say in this sentence.

Reply: Thank you for this comment. We agree and we made the correction to the text.

Remark 3: The paragraph from line 84 to line 87 describing the charge -3 defect is quite short, especially given the emphasis that is placed upon this result throughout your paper. It is true that the following paragraph gives some elaboration and there is no need to not be so succinct, but it is perhaps worth thinking about how brief the statement of this central result is.

Reply: We thank the referee for this important comment. We agree it is rather brief so we expanded it with a short comparison of the structure of the defect to a sp^3 orbital to help the reader build an intuitive picture of the defect.

Remark 4: I would like you to verify the direction assignment for arrows in your Figure 3. You write on line 48-49 that the orientation is outward from the droplet surface. On this basis I would assign precisely the opposite direction as you have done in Figure 3, so I would ask you to check to make sure you are correct.

Reply: Thank you for raising this issue. We were aware of this discrepancy in assignment of direction to the director, but unfortunately none of the two possible assignments served us in all situations. We assigned outward direction on the surface of the droplet as is common in literature and by this reproduced the signs of topological defects - radial/twisted hedgehogs with +1 topological charge and hyperbolic point defects with -1 charge. Unfortunately this assignment was unsuitable for Fig. 3, as it would invert the directions of arrows in Figs. 3a,b, changing the appearance of the ball with patches to a one with vectors on the blue "background" pointing outwards and the ones on red patches to point inward. This makes the image graphically very illegible so we decided to use the other possible assignment of vector direction, which shows the locations of patches better and makes the figure more intuitive in illustrating the patches as connection points for the surrounding defects. We found no better solution to this problem but we added a clear explanation of this fact in the caption to the figure and in the main text.

Remark 5: In line 126, I would advocate a moderately more reserved statement by removing the word "*possible*" from "... *minimum possible separation* ...".

Reply: Thank you for this comment. We agree and we made the correction to the text.

Remark 6: In lines 129-130 you write "*Our experiments in chiral nematic droplets clearly reveal, for the first time, the full anatomy of the hedgehog defects ...*". I think they do and I think this is an excellent achievement, and I think it is possible, by staring intently at the figures you have provided, to get a feeling for the structure of higher charge point defects, but it is not very easy. Part of this is to get a feeling for the nature of the chirality in these droplets. Minimization of the elastic energy will require that the director preferentially attains a configuration in which $\mathbf{n} \times (\nabla \times \mathbf{n}) = -q_0$ locally. The extent to which this is achieved (or, conversely, frustrated), and how it is, is not clear, to me at least, although I do appreciate that this is indicated by your 'cholesteric bubbles'. Nor is it immediately clear where the defects in the pitch axis (λ lines) are, or what their structure is like. It would be fascinating to figure all this out. I regret that I do not have anything useful to suggest to you, but still thought I would encourage you to continue thinking about such structures and promote your contribution.

Reply: Thank you for the comment and the encouragement for continuation of our work. These questions are of central interest to us, because they present a path to understanding complex 3D structures in terms of more basic building blocks. Unfortunately there are considerable obstacles in such

analysis of experimental data. In particular, numerical calculation of the derivatives in $\mathbf{n} \times (\nabla \times \mathbf{n})$ from experimentally reconstructed data is poorly conditioned and shows a lot of noise and artefacts. This time we did not present these data, but something similar has been shown in our previous paper on data of sufficient signal to noise ratio [30].

As you noted we tried to supplement such analysis with the notion of cholesteric bubbles. The presented structures are created from a quench, which creates a lot of defect pairs and loops that contract into points. The homeotropic boundary condition in fact enforces $\mathbf{n} \times (\nabla \times \mathbf{n}) = 0$, which is directly incompatible with the cholesteric condition. Similarly, chirality is almost zero around point defects. The cholesteric bubbles are the regions where the cholesteric twist is localized in our very frustrated systems - the director in these regions tries to minimize the term $\mathbf{n} \times (\nabla \times \mathbf{n}) - q_0$ by forming structures related to double-twist cylinders and closely connected to skyrmions and torons. Conversely if we reduce the frustration by making the droplet much larger compared to the pitch, or if we cool down gradually instead of quenching, a cholesteric ground state can be established in most of the droplet, with the exception of the near-surface region. We have captured many 3D images and only a portion of these are analyzed in detail. The rest will be prepared for publication in the near future.

Remark 7: As a final comment I might say that one of the striking features of your experiments is that +1 defects are attracted to the surface of the droplet, while negative charge defects are pushed to the interior. In a nematic droplet, the simplest configuration one can think of has a single +1 defect at the centre; it is not attracted to the surface. One may reasonably ask why the cholesteric droplets do not simply produce a chiral distortion of this configuration, especially when the pitch is comparable to the droplet size. Accounting for this contrast seems to me to be a first challenge for theory.

Reply: This is indeed an interesting question. The argument from our previous answer to Reviewer #2 also explains this one: in a cholesteric, the surface enforces a non-cholesteric layer, and the cholesteric can only form in the bulk (either as bubbles, or, in larger droplets, as more layer-like structures). Point defects prevent significant twisting and are therefore expelled toward the surface to maximize the volume available for cholesteric structures. In a nematic, there is no requirement for cholestericity, and therefore no tendency for expelling the defects to make room for the cholesteric.

The negatively charged defects in our structures cannot be expelled and annihilated with their corresponding surface partners due to the cholesteric bubble between them - or rather, if there were pairs created without a bubble between them, they did not survive the quenching, and only those that did, remained. We expanded the discussion in our paper with an additional paragraph to clarify this matter.

REVIEWERS' COMMENTS:

Reviewer #1 (Remarks to the Author):

All the comment were properly addressed and now the revised manuscript seems to be more persuasive and well-organized to explain the details. And thus, I recommend publication of this work in Nat. Comm. without hesitation.

Reviewer #2 (Remarks to the Author):

In the manuscript "Hidden topological constellations and polyvalent charges in chiral nematic droplets" the authors have addressed my comment and improved the text. I recommend the manuscript for publication in Nature Communications.

Reviewer #3 (Remarks to the Author):

The authors have carefully addressed all comments from the three referees and made suitable changes and improvements to their manuscript in response. I reiterate what I said previously: I believe the paper should be accepted for publication in Nature Communications and recommend that it can be accepted in the present form.